# Outcomes of Open Repair Treatment for Acute Versus Chronic Achilles Tendon Ruptures: Long-Term Retrospective Follow-Up of a Minimum 10 Years—A Pilot Study

**DOI:** 10.3390/medsci11020025

**Published:** 2023-03-24

**Authors:** Marta Tarczyńska, Mateusz Szubstarski, Krzysztof Gawęda, Piotr Przybylski, Elżbieta Czekajska-Chehab

**Affiliations:** 1Orthopaedic Surgery and Traumatology Department, Medical University of Lublin, 20-950 Lublin, Poland; 2Orthopaedic and Traumatology Department, District Hospital of Krasnik, 20-950 Lublin, Poland; 31st Department of Medical Radiology, Medical University of Lublin, 20-950 Lublin, Poland

**Keywords:** calcaneal tendon, rupture, delayed treatment

## Abstract

The aim of the present study was to compare repaired Achilles tendon (AT) remodelling, whether its function was restored and what effects the surgery had on our patients’ gait cycle in a long-term follow-up study. The study population comprised 30 human subjects treated acutely and chronically for AT ruptures, using the same surgical technique in all cases. The study group was divided into two subgroups regarding the age of their AT injury, i.e., how much time elapsed between the injury and when a correct diagnosis was made and when adequate treatment was applied. Following these criteria, persons presenting at less than 4 weeks postinjury were classified as acute rupture (AR) patients and those presenting at more than 4 weeks after injury were grouped as chronic rupture (CR) patients. Both patient groups were operated on using a surgical method favoured at least a decade ago, i.e., open repair through a posteromedial approach. The AT was augmented with a plantaris longus tendon autograft, followed by suturing using the pull-out suture technique. The results were measured using clinical, ultrasonographic (US) and pedobarographic methods. Our ultrasonographic and pedobarographic findings revealed differences between both patient groups, thus indicating that delayed surgery had negative impacts on treatment success, however, with good long-term functional score outcomes in both patient groups. Nevertheless, delayed treatment of AT ruptures did not leave individual gait phases unaffected, as it also affected the plantar surface and balance performance of the affected limb. As per the results, the Achilles tendon manifested decreased capacity following delayed treatment; however, its long-term functional outcomes were favourable, irrespective of whether it was for acute or chronic patients.

## 1. Introduction

The Achilles tendon (AT) is the largest and strongest tendon in the human body [1]. In being so, it is frequently ruptured, as it is repeatedly overloaded during daily activities [2,3]. According to some publications, Achilles tendon ruptures account for 30 to 50% of all sports-related damage to tendon structures [4]. There are many protocols for AT injury treatment [5,6]. AT ruptures are particularly demanding to repair in chronic conditions [7,8]. According to some authors, the incidence of this type of AT rupture can be as high as 20% of all AT injuries [9]. The delayed treatment of AT ruptures is usually attributable to either poor diagnosis upon injury or patients disregarding their injuries and not presenting to healthcare facilities. Although there are numerous reports on AT injuries and their treatment [5,7,10,11,12,13], there are no studies assessing the impacts of delayed diagnosis and treatment on mid- and long-term outcomes [14,15]. Therefore, more research in this area is needed to establish the effects of delayed proper diagnosis and treatment on patient functional outcomes, gait biomechanics and ultrasonographic (US) appearance of the repaired Achilles tendon during its remodelling.

The aim of our study was to retrospectively evaluate the effects of delayed treatment of AT injuries on clinical outcomes of operatively treated tendons, on their ultrasonographic features and on gait biomechanics in a long-term follow-up study.

## 2. Methods

Our study population comprised 30 patients treated operatively for AT injuries in the years 1996–2011. In order to qualify for the study, our patients had to meet the following criteria: isolated traumatic injury to the Achilles tendon of one limb, treatment using the same surgical technique and the same rehabilitation protocol in all cases, as well as informed consent to participate in a retrospective study of a minimum of 10 years after surgery. The study population was divided into two subgroups regarding the age of their Achilles tendon injury, i.e., as to how much time elapsed between the injury and when a correct diagnosis was made and adequate treatment was applied. As per these criteria, 15 persons operated on at less than 4 weeks after injury (mean: 3 days, range: 1–18 days) were grouped as acute rupture (AR) patients, and 15 persons operated on at more than 4 weeks after injury were classified as chronic rupture (CR) patients (mean: 80 days, range: 30–180 days).

The age range at injury was 34–53 years for AR patients (mean: 43.6 years) and 23–59 years for CR patients (mean: 44.4 years). During follow-up, both patient groups were also examined for the presence of other unrelated foot and shin conditions. None of them self-reported or exhibited any clinical features of other such health conditions. All individuals in both study groups self-reported recreational levels of pre-injury physical activity and did not self-report any history of chronic disease treatment. In the CR group, there were occasional cases of NSAID or analgesic use for exacerbated pain management in the injured area after injury and prior to definitive treatment of the damaged tendon.

Following the repair and prior to the last follow-up examination, our patients were neither observed to suffer any other damage to the limb with the repaired Achilles tendon nor did they undergo any other surgical procedures on the affected limb.

In the AR group, 81% of the patients (12 persons) were male and 19% (3 persons) were female. The CR group consisted of 60% male patients (9 persons) and 40% female patients (6 persons).

In all patients, the same surgical method was used—favoured in those years, i.e., open repair using a posteromedial incision. The tendons were repaired using a Kessler suture augmented with plantaris longus tendon autografts. In all patients, tension in the repair area was reduced using the pull-out suture technique. To remove myotendinous adhesions, almost all CR patients had to have a nonextensive tenolysis of the distal and proximal stumps of the Achilles tendon performed.

In order to reduce passive tension on the healing tendon and prevent re-rupture, the operated limb was immobilised in a plaster cast, with the foot flexed at approximately 30 degrees of plantar flexion. The patients were mobilised nonweight bearing with two elbow crutches. At four weeks, the crutches were discontinued, and the plaster cast was replaced with a walker. In addition, passive plantar flexion and dorsiflexion exercises of the affected foot were initiated. After another three weeks, active exercises were commenced, and the patients were permitted to walk with one elbow crutch. After another three weeks, full weight bearing without crutches was allowed. The rehabilitation protocol was supplemented with sport-specific training, including swimming pool activities and cycling, as well as closed kinetic chain proprioception exercises. Two weeks later, open-chain proprioception exercises and gym activities were introduced. At four months after repair, the patients returned to full activity.

When qualifying patients for our outcome evaluation of AT rupture treatment, the following criteria were taken into account: AT open repair through a posteromedial incision, time between injury and surgery and a follow-up period of at least 10 years. The mean follow-up period was 13 years (range: 10–19 years) for AR patients and 17 years (range: 10–25 years) for CR patients. In both groups, contralateral normal limbs were used as controls. The study was approved by the local Bioethics Committee (no: KE-0254/203/2017), and all subjects gave their voluntary, informed consent to participate.

### 2.1. Clinical Scales

Mid- and long-term outcomes of open repair treatment for AT ruptures were evaluated using the American Orthopaedic Foot and Ankle Society (AOFAS) Ankle-Hindfoot Scale [16], the Victorian Institute of Sports Assessment-Achilles (VISA-A) questionnaire [17] and the Visual Analogue Scale (VAS) for patient satisfaction [18]. The outcome scoring scales were used in their Polish validated versions. When the research was initiated, the only validated scale in Polish was the AOFAS for the hindfoot, with the ATRS scale not yet validated at that time.

### 2.2. Ultrasonographic Assessment

Ultrasonography (US) is well documented as a reliable and sensitive imaging method to assess the morphology and healing of tendon lesions managed with different repair techniques and rehabilitation protocols [19,20]. In this study, US assessment was carried out using a MyLab^TM^ Class C scanner (Esaote, Italy) with standard convex (2–5 MHz frequency range) and linear (4–12 MHz frequency range) probes.

In all patients, US examinations were performed by a radiologist specialising in musculoskeletal imaging, who was unaware of which patient was assigned to which group.

The repaired AT, the Kager’s triangle, and the adjacent bursae were visualised using B-mode US and power-Doppler ultrasonography (PDUS). B-mode US was used to determine (1) tendon echogenicity, (2) abnormalities in the architecture of tendon fibres, along with the distance of the lesion from the calcaneal enthesis, (3) thickness of the repaired tendon in the sagittal and transverse planes and (4) thickness of the paratenon. In ultrasound examination, tendon peritoneum enlargement over the repair area was assessed against its thickness in the healthy limb, measured at the same distance from the top of the calcaneal tuberosity. Identically, the thickening of the tendon itself at the site of its repair was examined. On two longitudinal images with the probe on the posterior and lateral surface of the tendon, its linear dimension was measured at the repair site. The extent of the thickening was indicated by the difference between the measurements in the operated tendon area and the dimensions of the healthy tendon in the nonoperated limb at an identical distance to the repair area in the operated tendon. Tendon mobility was compared by measuring in cm the distance by which the central part of the repaired area was moved when the movement was performed from maximum active sole flexion to maximum dorsiflexion of the foot. The linear measurement obtained was compared in an identical manner with the movement of a part of the healthy tendon located at an identical distance to the calcaneal tuberosity as the repair area in the affected limb.

In addition, the insertion of the AT into the calcaneus was evaluated for signs of mineralising enthesopathy, and passive and active movement of the tendon within the surrounding tissues was assessed. The quality of blood flow to the AT and the surrounding tissues was evaluated using PDUS.

### 2.3. Functional Assessment

A FreeMED BASE pedobarographic platform (Koordynacja, Poland) was used to test the patients’ gait biomechanics. Both static and dynamic measurements were performed.

The static examination was carried out to determine the plantar pressure distribution (i.e., location and area of plantar pressure centres). Particular attention was paid to the distribution of pressure in different regions of the sole and of peak plantar pressure during single- and double-leg stance. The test took place on a 40 × 40 cm one-piece mat, to which 2 pieces of passive plantar were connected to align the gait path. The platform was located 2 m from one wall of the room. Once the platform was connected to the power supply, calibration was performed before each test. The assessment began with a static test, which involved positioning the patient on the platform with their feet in specific sensory fields of the device with their face to the wall. In the first stage of the assessment, the patient took a few steps in place in order to familiarise themselves with the surface and normalise their stance. After stabilising the stance, the patient stood still on the mat, at which point the data acquisition of the force distribution on the sole surface of the feet was switched on for 5 s with the eyes open, and for another 5 s the acquisition was performed when the patient’s eyes were closed. In the second stage, the test was performed for 10 s, with the patient standing on only one unoperated limb with their eyes open as previously in the first part of the assessment and with their eyes closed in the second part. In the final stage, an identical procedure was performed while the patient was standing on the operated limb. Throughout the assessment, the real-time data received were processed using FreeStep software integrated with the platform. At the end of the static part of the assessment, we obtained the following data: load distribution and area of both limbs, precise distribution of pressure ratios, area of pressure and maximum forces acting on the feet; in the balance test: swing length and ellipse area.

The dynamic tests were used to measure pressure distribution in different parts of the sole in the individual gait phases. We determined the forces acting on the individual regions of the foot and parameters such as duration of push-off and average foot rotation.

In the second dynamic stage of the assessment, the procedure was started by positioning the patient in one corner of the passive pathway. During the preparatory phase, the patient was instructed to march with their natural step of a normal gait to the opposite corner of the second passive plantar. Once the gait was normalised and stabilised, the assessment phase began. The patient took 20 to 25 steps, during which the real-time data were recorded. In the dynamic test, the following parameters were analysed: distribution of foot pressures in the individual gait phases and magnitude of the forces acting on the individual foot areas, as well as push-off length and mean foot rotation.

We also conducted a postural balance test to measure postural sway and interaction of the affected foot with the supporting surface, with the patients standing on the injured limb for 10 s with eyes open or closed. Both the sway path and the centre of pressure sway area were calculated.

The results obtained were analysed using FreeStep software integrated with the pressure platform.

### 2.4. Statistical Analysis

The results were analysed statistically using STATISTICA 10.0PL software (StatSoft Inc., Tulusa, OK, USA) [21].

Because the measurements were highly skewed, they are presented as the median, a measure of central tendency. The normality of distribution of the individual variables within both groups was tested using the Lilliefors test (a version of the Kolmogorov–Smirnov test) and the Shapiro–Wilk test. Since the test variables did not have a normal distribution, nonparametric tests were used in further analyses. These tests are robust to deviations from normality of distribution and heterogeneity of variance.

The Mann–Whitney U test was performed to compare differences in interval variables between the two independent groups. Differences in nominal variables between the two independent groups were tested using a chi-square test.

## 3. Results

### 3.1. Clinical Scales

Treatment outcomes, assessed using the dedicated clinical scales, were good and very good for both AR and CR patients. The mean AOFAS score was 91.2 points (range: 68–100) for AR patients and 89.4 points (range: 70–100) for CR patients.

The VISA-A scores of the AR patients ranged between 75 and 97 points, with a mean score of 88 points, and those of the CR patients ranged from 75 to 100 points, with a mean score of 86.7 points.

High scores were obtained on the patient satisfaction VAS. The mean VAS score was 88.3 in both groups. The score ranges were 65–100 for the AR patients and 75–100 for the CR patients. The results are presented in Table 1.

### 3.2. Ultrasonographic Assessment

In both patient groups, the repaired tendons showed expected US changes in comparison to the control tendons. The findings included reduced echogenicity of the repair area, loss of normal fibrillary echotexture, tendon thickening in the sagittal plane and restricted tendon movements within the surrounding tissues. These findings did not differ significantly between both patient groups (χ^2^ = 0.00; df = 1; n.i.) (Table 2).

When US images of the repair areas were compared between both groups, differences in the morphology of tendon repair were observed. Thickening of the paratenon was recorded in the AR patients (χ^2^ = 5.40; df = 1; *p* = 0.02) (Table 2). In this patient group, the lesions were located more distally, closer to the tendon insertion into the calcaneus, compared to the CR group. On the other hand, patients in the CR group were more likely to have US signs of Achilles tendon enthesopathy (χ^2^ = 6.65; df = 1; *p* = 0.01) (Table 2).

### 3.3. Functional Assessment

Changes in the morphology of the plantar surface of the affected foot and gait pattern abnormalities were examined in the static and dynamic pedobarographic evaluations. Both studies showed differences between the AR and CR patients.

In the static study, the CR patients were found to have greater plantar pressure under the midfoot (*p* = 0.049) in comparison to the AR patients. In the dynamic study, the CR patients demonstrated significantly lower peak plantar pressure (*p* = 0.044), longer heel strike (*p* = 0.021) and midstance (*p* = 0.007) phases and an increase in pressure area during midstance (*p* = 0.003) and push-off (*p* = 0.034) when compared to the AR patients.

The pedobarographic evaluations also included an analysis of balance. In a balance task in which the affected limb was loaded, the CR patients had a significantly larger centre of pressure sway area in the both eyes-open (*p* = 0.004) and eyes-closed (*p* = 0.008) conditions. Similar differences between the CR and AR patients were found for sway path during loading of the affected limb (*p* = 0.003). The results obtained in the conducted research are shown in Table 3.

## 4. Discussion

The AT plays an important static and dynamic roles in maintaining postural balance during motor activity. For this reason, treatment of AT injuries requires detailed and comprehensive research [22,23]. According to the literature, open repair, which used to be the most commonly employed operative treatment method before the introduction of minimally invasive percutaneous repair techniques, is often associated with postoperative wound complications [24]. Despite its drawbacks, however, this method has been repeatedly reported to provide good clinical outcomes [25]. Our observations also confirmed the efficacy of this treatment approach. All operated patients achieved good and very good scores on the clinical scales used. There were also no postoperative wound complications.

Because chronic AT ruptures are not common complaints in daily clinical practice [9], they pose a challenge to surgical teams, who often have little experience in managing this type of injury, all the more so that there are no straightforward, commonly accepted clinical management algorithms or rehabilitation protocols for postoperative follow-up management of missed or neglected AT ruptures [26].

In a prospective study, Yasuda et al. analysed the treatment outcomes of 30 patients whose injured ATs were repaired more than at four weeks after injury [27]. The surgical approach they used to restore the continuity of the tendons was comparable to the one reported in this present study. Their human subjects were able to perform daily activities and practise various recreational sports without any restrictions postoperatively. In all patients, the operative treatment significantly improved their AOFAS scores [15].

It should be emphasised that the relationship between the time from injury to surgery and the treatment outcomes has been investigated in only a few clinical trials. We conducted a search of scientific databases which returned only two relevant citations [13,14].

In 2018, Becher et al. reported on a study of a cohort of AT rupture patients similar in size to our sample [13]. In that study, which included a group of 30 patients with an average follow-up time of less than 5 years, the authors compared the outcomes of operative treatment in patients with acute versus chronic AT ruptures. Their patients with acute lesions were treated with mini-open repair (i.e., percutaneous suturing), and those with chronic injuries received standard open surgery. Because the authors used various open reconstruction techniques (both standard and with autograft augmentation) in their CR patients, their sample was not uniform, and so only a tentative comparison can be made with our study [13].

Becher et al.’s clinical and imaging (US and MRI) findings were similar to ours. They showed that both AR and CR patients achieved good and very good clinical outcomes, as measured using the AT Total Rupture Score (ATRS), VISA-A and VAS. Of note is the fact that the authors did not use the AOFAS scale, which is employed in most other reports in this research area [13].

In Becher et al.’s study, imaging of the repaired tendons did not show any significant differences between their AR and CR patients, except for significantly less mediolateral tendon thickening in the former. The researchers noted, however, that these observations did not correlate with their clinical or functional findings [13]. Our imaging results are fully consistent with those reported by Becher et al. The US did not reveal any significant differences in the structure of the repaired AT between our AR and CR patients. We only observed thickening of the tendon in the AR group, which we believe may be related to the location of the injury area. In our AR patients, the lesions were located more distally, closer to the tendon enthesis of the tuber calcanei, as compared to our CR patients. Contrary to our findings, Becher et al. did not observe a higher prevalence of AT enthesopathy in their patients who had undergone delayed surgery [13]. In our study, enthesopathy may have developed in the CR patients, as they tried to compensate for the reduced elasticity of the scar around the AT repair area, which was far larger than in the AR patients. The difference between our findings and those obtained by Becher et al. may also be due to the much longer postoperative follow-up period in our study. With time, tendon function in our CR patients deteriorated and secondary changes developed.

Another study that analysed the impacts of delayed diagnosis and repair on the treatment outcome of AT ruptures was reported on in a paper by Carmont et al., also published in 2018 [14]. The authors compared patients with early versus late repair of AT ruptures. In their patients with delayed presentation, the time from injury to surgery was more than 14 days (mean: 21.8 days). The patients who received delayed treatment with minimally invasive open repair treatment had similar postoperative functional scores to those treated acutely. According to Carmont et al., their findings demonstrated that a delay in diagnosis and treatment did not significantly affect the clinical outcomes [14]. One weakness of that study was that the period of 14 days from AT rupture to its treatment did not meet the commonly accepted cut-off for chronic injury [9]. The short- and medium-term follow-up periods are another limitation of that report.

In our study, as part of the pedobarographic examination, we assessed the patients’ postural balance. In a balance task in which the repaired extremity was loaded, our CR patients had a significantly larger centre of pressure sway area, in the both eyes-open and eyes-closed conditions, compared to our AR patients. Our CR patients also exhibited a greater sway path length than the AR subjects when weight was borne on the affected limb. This may indicate that the time from injury to treatment may affect the function of the AT measured as the ability to neutralise postural sway. In the pedobarographic examination, we also observed a change in the morphology of the plantar surface of the hindfoot and lengthening of the individual gait phases in the CR patients. These findings may be indicative of a decrease in the function of the CR tendons, which may lead to sustained impairment in hindfoot weight-bearing, progressing gait asymmetry, gradual development of degenerative changes in the tarsal joints and enthesopathic changes in the AT.

Since we did not identify any studies in the available scientific databases that assessed the effects of delayed AT treatment on gait performance, we cannot compare our results with other reports.

Our findings demonstrate that delayed treatment does not significantly affect patients’ functional outcomes, measured with the available clinical scales, or their self-reported satisfaction. However, the abnormalities detected in US scans and pedobarographic measurements showed that these patients should be evaluated regularly to determine the durability of the clinical benefit obtained.

It should be emphasised that the differences between the AR and CR patients seen in the US and pedobarographic examinations may affect their daily functional performance of the affected limb in the future, especially in the latter group of patients. Delayed treatment of the AT has an influence on patient gait phases, plantar pressure area and performance of the affected limb in balance tests. The pedobarographic findings are consistent with the US findings. Therefore, it seems important to highlight the need for health providers to carry out thorough physical examinations of AT injuries with the skilful use of imaging techniques so that there are fewer cases of delayed diagnosis.

The weakness of this work is the small number of patients. However, after analysing the available literature, our study groups, unlike in other publications, are homogeneous and as numerous as or more numerous than in other available studies, as published on such long-term assessments of calcaneal tendon repair. A lack of concentration of the results in some of our assessments (e.g., in the baropodometry test) may reduce the reliability of the results obtained. Further investigation of the impacts of delayed surgery on patient treatment outcomes is needed, especially with regard to mid- and long-term outcomes.

## 5. Conclusions

The delayed treatment of a ruptured AT results in the lengthening of some gait phases and changes in the shape and size of plantar pressure areas, as well as affects the balance performance of the affected limb.

Despite the differences in our US and pedobarographic findings indicating that delayed surgery had negative impacts on patient treatment outcomes, the mid- and long-term functional outcomes of AT ruptures were good in both acute and chronic rupture patients.

## Figures and Tables

**Table 1 medsci-11-00025-t001:** Clinical scales results.

	Chronic	Acute	Z	*p*
	Median	Median		
VISA-A	90	90	−0.44	0.663
AOFAS	90	95	−0.60	0.548
VAS	90	90	−0.12	0.901

**Table 2 medsci-11-00025-t002:** Ultrasonographic assessment.

	Chronic Injury		Acute Injury	
Echogenicity of the affected tendon	N	%	N	%
below 50%	7	46.67	7	46.67
above 50%	8	53.33	8	53.33
χ^2^ = 0.00; df = 1; n.i.				
Fibrillary echotexture of the affected tendon	N	%	N	%
below 50%	6	40.00	6	40.00
above 50%	9	60.00	9	60.00
χ^2^ = 0.00; df = 1; n.i.				
Dimensional enlargement of the tendon peritoneum	N	%	N	%
normal	13	86.67	7	46.6
thickened	2	13.33	8	53.33
χ^2^ = 5.40; df = 1; *p* = 0.02				
Tendon thickening in the sagittal plane	N	%	N	%
no	6	40.00	4	26.67
yes	9	60.00	11	73.33
χ^2^ = 0.60; df = 1; n.i.				
Tendon mobility within the surrounding tissues	N	%	N	%
normal	2	13.33	2	13.33
abnormal	13	86.67	13	86.67
χ^2^ = 0.00; df = 1; n.i.				
Achilles tendon enthesopathy	N	%	N	%
no	5	33.33	12	80.00
yes	10	66.67	3	20.00
χ^2^ = 6.65; df = 1; *p* = 0.01				

**Table 3 medsci-11-00025-t003:** Functional assessment.

	Chronic Injury	Acute Injury	Z	*p*
	Median Value	Median Value		
Sway path during loading of the affected limb (mm)	786.69	395	2.92	0.003
Centre of pressure sway area in eyes-open conditions (mm^2^)	5000	887	2.86	0.004
Centre of pressure sway area in eyes-closed conditions (mm^2^)	1595	600	2.65	0.008
Plantar pressure under the midfoot (%)	4.49	2.3	1.97	0.049
Peak plantar pressure (gr/cm^2^)	1160	1395	−2.01	0.044
Heel strike phase (ms)	37	31	2.30	0.021
Midstance phase (ms)	410	324	2.68	0.007
Pressure area during midstance phase (cm^2^)	150	139	2.94	0.003
Pressure area during push-off phase (cm^2^)	144	135	2.12	0.034

## Data Availability

The data presented in this study are available on request from the corresponding author. The data are not publicly available due to human subjects.

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
