# Peer review of "Outcomes of Open Repair Treatment for Acute Versus Chronic Achilles Tendon Ruptures: Long-Term Retrospective Follow-Up of a Minimum 10 Years—A Pilot Study"

_medsci, 2023, doi:10.3390/medsci11020025_

Round 1

Reviewer 1 Report

1. The criteria for ultrasonographic assessment should be clearly provided especially tendon peritoneum enlargement, tendon thickening and tendon mobility.

2. The methods of foot pressure and balance test can be more explained for authors.

Author Response

Thank you very much for constructive suggestions. We changed the text in accordance to them. The changes are in the attached file. 

Reviewer 2 Report

On the results, the part where the mid- and long-term functional outcomes of AT rupture are good in both acute and chronic rupture patients should be increased

Author Response

Dear Reviewer,

 thank you for your suggestions.

The changes in accordance to your comments were done are listed in the attachement.

Reviewer 3 Report

Unfortunately while reviewing your manuscript I found major concerns.
The title is inappropriate and written in a bad english. Overall the whole manuscript needs a extensive english proof-reading. Also the structure of the whole paper is not appropriate. I would recommend the authors to check for a lecture in scientific writing. The abstract of the manuscript is not structured and lacks in scientific matters.

In the current form and with this major concerns I do not see any possibility of publication of your report. I would suggest you to do a major revision and extensive proof-reading of your manuscript. Then a renewed submission process can be started.  

Author Response

Dear Reviewer 3, 

the answer is in the attacement.

Reviewer 4 Report

Dear Authors,

    First of all, I think the manuscript entitled: “Outcomes of acute versus chronic open repair Achilles tendon rupture. Long term retrospective follow up, minimum 10 years.” submitted for publication in Medical Sciences Journal (MDPI) has a scientific interest.

More specifically:

·         The aim of the present study was to compare the remodelling and function of the repaired Achilles tendon (AT) and the effect of the surgery on gait performance in acutely and chronically treated patients in long-term follow-up.

·         The manuscript text needs correction (in several places of the text, there are deletions of part of the text as well as additions in red color text).

·         The manuscript's conclusions are in accordance with the evidence and the arguments presented by the authors.

·         The authors address the central question quite well.

Based on the above:

Overall Recommendation: Revision.

Comments and Suggestions for Authors:

1) The manuscript text needs correction (in several places of the text, there are deletions of part of the text as well as additions in red color text).

2) As mentioned in the manuscript (lines 53-54), the sample of this study consisted of 30 patients with Achilles Tendon (AT) injury. Did the researchers perform any procedure to determine the necessary sample size of the study (as well as sample size calculation);

If yes, I think it should be mentioned in the materials and methods chapter.

If not, I would suggest that the title of the study be changed to: “Outcomes of acute versus chronic open repair Achilles tendon rupture. Long-term retrospective follow-up, minimum 10 years: A pilot study.”

Author Response

Dear Reviewer 4,

thank you very much for your suggestions. The answeres to all of them are in the attachement. We hope that text changes are satisfied.
